# The Push and Pull of Network Mobility: How Those High in Trait-Level Neuroticism Can Come to Occupy Peripheral Network Positions

**DOI:** 10.3390/bs9070069

**Published:** 2019-06-28

**Authors:** Eric Gladstone, Kathleen M. O’Connor, Wyatt Taylor

**Affiliations:** 1Institute for Analytical Sociology, LINKS Center for Social Network Analysis, Gatton College of Business and Economics, University of Kentucky, Lexington, KY 40507, USA; 2London Business School, University of London, London NW1 4SA, UK; 3Gatton College of Business and Economics, University of Kentucky, Lexington, KY 40506, USA

**Keywords:** social networks, individual differences, traits, neuroticism, emotional stability, network evolution, experiment

## Abstract

Field research shows that people’s network positions are determined, at least in part, by their traits. For instance, over time, actors higher in trait-level neuroticism move out to the network periphery. What is unknown is how this happens. Drawing on personality and social psychological theory, we generated a model that could explain the movement of actors who are higher in neuroticism. Our aim is to add to the existing empirical literature on the interplay of actor level traits and social networks, and do so using methods that can establish possible causal pathways. In four experiments, we tested two explanatory mechanisms—aversion on the part of alters and avoidance on the part of focal actors. Results showed that potential alters indeed perceived actors higher in neuroticism as aversive, leading them to block these actors from well-connected spots. Specifically, low perceived levels of likability prevented actors from being nominated to better positions. In a test of avoidance, actors higher in neuroticism recognized the benefits of better-connected network positions, but also saw them as costly, and thus, declined opportunities to occupy them. This work shows how both alters and egos can determine egos’ place in networks, and specifies how this is done.

## 1. The Push and Pull of Network Mobility: Introduction

The advent of increasingly sophisticated analytic and graphing tools has made tracking network dynamics–and especially shifts in the shape and density of networks–ever easier [1,2,3,4] While recognizing and cataloging patterns of tie creation, deepening, and decay is important, these efforts raise, but often leave unanswered, questions about the origins and causes of these dynamics. Are there traits, or qualities, of actors that may help us understand who comes to occupy the network positions that they do? If we wish to continue deepening our understanding of how and why certain classes of actors occupy certain type of network positions, particularly in workplace networks, and how these relationships change over time, investigations should center on questions of “how” as well as “whom”.

Starting with the question of “who goes where?” scholars have investigated links between actors’ personality traits and their network positions, and they have done so both in snapshot studies and in studies relying on longitudinal designs. Studies show, for instance, that people who score highly on measures of self-monitoring are relatively more likely to locate in central positions that deliver material and professional benefits [5], as well as being more likely to move towards advantageous positions over time [6]. In a longitudinal study of network movement over a three-month period, [7] also investigated the impact of a wide range of demographic and trait characteristics on network mobility. Among others, they found that actors who scored higher in trait-level neuroticism were more likely to move from more to less central positions in their advice and friendship networks.

Answers to the question of “who moves” are piling up, while answers to the question of how moves actually come about—that is, accounts of the decisions and actions that account for mobility—have lagged. Failure to empirically unpack the mechanisms that could move actors from one position to another means that our ability to predict network mobility is hampered; on a theoretical level, it means that our ability to develop a theory of network evolution that incorporates human cognition and trait characteristics is impaired. To contribute to theory development, we identify and test a set of cognitive and behavioral pathways that can explain how actors with certain traits—here, trait-level neuroticism—come to occupy particular network positions. Moreover, we consider the critical roles both egos (i.e., focal actors) and alters (i.e., tie partners) may play in determining egos’ network positions.

Our specific point of departure is the established link between actors’ trait-level neuroticism and their peripheral network positions [7]. Not only does neuroticism carry network consequences, as this work shows, but it continues to garner empirical attention from personality and social psychologists as well as neuroscientists [8,9,10,11]. Thus, the cognitive and behavioral correlates of neuroticism are well understood, providing us with a foundation for making strong theoretical predictions.

Below, we develop and test a conceptual model that describes a set of individual-level processes that may explain why and how actors who are higher in neuroticism (HN actors) can become systematically less central over time, as field studies have shown. According to personality theorists who work with the Big Five [12], each of the five traits is a dimension that runs from low to high. Hence, it is most appropriate to refer to people as scoring lower or higher on a trait dimension. However, to ease readability, we will use ‘HN’ to stand in for ‘higher in trait-level neuroticism’ throughout the paper. While focusing on individual-level cognitions and behaviors, our model acknowledges networks as comprising pairs in which either person can choose to initiate a tie or not. This means that actors’ neuroticism can affect the decisions and behaviors of both alters and egos. Finally, using experimental methods we control for the effects of other network-relevant traits, such as extraversion and self-monitoring, to focus exclusively on explaining how neuroticism can move actors through their networks. These methods allow us to shed additional causal light on the empirical processes identified by other researchers.

## 2. Neuroticism and Networks

In their work documenting the impact of actor traits on actor network position, Klein and her colleagues [7] offered a handful of hypotheses around how actors’ Big Five personality traits—extraversion, agreeableness, openness to experience, conscientiousness, and neuroticism— [12] might affect their position in their networks. Their findings showed that, relative to other traits, neuroticism proved to be the strongest predictor of network position in the advice, friendship, and adversarial networks studied. Specifically, individuals who were higher in trait-level neuroticism stood a lower chance of being central in their advice and friendship networks but were relatively more likely to be central in their adversarial networks. In the absence of causal or descriptive data, the authors speculated that actors who score highly on neuroticism measures are unpleasant interaction partners. Unpleasant interaction partners are unlikely to be targeted for ties, the argument went, making them more likely to be moved to the periphery. Indeed, empirical research supports the notion that actors high in neuroticism are likely to make unappealing prospects for ties [13,14,15,16].

HN actors themselves may bear responsibility for their network spots. People who score highly on neuroticism measures are characterized by negative emotional states, including anger, depression, and frustration [12,17] Compared to others, these individuals are especially vulnerable to experiencing anxiety and distress. In fact, sensitivity to threat and anxiety appear to form the emotional core of neuroticism as a trait [14,15]. What is more, this tendency to experience anxiety is even more pronounced in the face of uncertainty [18] Neurophysiological research demonstrates this quite clearly, with studies showing that those individuals higher in neuroticism respond more strongly to uncertain than to negative feedback [19].

The tendency of HN actors to experience negative emotions may cause them to behave in ways that are relevant to understanding social networks. Actors’ trait-level neuroticism scores are significantly and positively related to quarrelsome interpersonal behavior, which includes confronting others and making sarcastic comments, for instance [20,21,22]. In online social fora, neurotics tend to post longer, more strongly worded posts, which tend to lead to more alter comments and support [23]. Conversely, neuroticism is negatively related to expressions of positivity, such as smiling and laughing, and expressing affection. Meta-analytic work reveals that people who are relatively high in neuroticism are more likely to disengage from others and to behave passively in their relationships [24,25]. That is, they are more likely than others to retreat from interpersonal interactions and avoid situations requiring high degrees of interpersonal engagement. This is likely to explain why neuroticism exerts stronger effects on closer ties than on on those more distant [26].

Taken together, these findings indicate that potential alters have reason to overlook people who score high in neuroticism. Meta-analytic work has confirmed that HN actors tend to show lower indegree values than others [27] at any given time, though there are no differences in brokerage levels. Thus, whether neurotic people will be left aside in favor of others, thereby shifting them to the perimeters of their networks is an open empirical question. Moreover, besides the aversion alters are likely to feel toward egos high in neuroticism, our review indicates a second mobility mechanism that has yet to be identified and tested. Given that their interpersonal interactions are marked by friction and difficulty, HN actors may find positions that are structurally well connected, or that demand high levels of interpersonal contact, relatively less appealing. Thus, actors themselves may eschew opportunities to occupy better connected or more central spots due to perceived costs.

We posit two principal mechanisms that can explain how actors who are higher in trait-level neuroticism might move to the edges of their networks over time, as Klein and her colleagues [7] found: alters’ aversion and egos’ avoidance in Figure 1.

Relying on experimental methods, we put both mechanisms to the test to investigate the possible causal role of actors’ trait-level neuroticism in social network evolution and to illuminate mechanisms by which social networks may change. Because people’s interactions with actors who are higher in neuroticism are likely to be unpleasant, people will prefer not to interact with them. In addition, the false consensus effect [28,29,30] suggests that people will assume that others hold the same preferences. We will call this the *aversion* hypothesis: potential alters are averse to interacting with people high in neuroticism and believe that others will be as well, leaving these actors with relatively fewer opportunities to move into more central positions in their networks. In two experiments we test this hypothesis.

The second mechanism we posit is *avoidance*. Given their tendency to experience friction and stress in their interpersonal interactions, neuroticism may mean some actors will have relatively low interest in better-connected network positions. By their own decisions and actions, then, these actors will move to the periphery. In our third and fourth studies, we test whether neuroticism affects actors’ preferences for and choice more central or less central network positions. Results of these four experiments will illustrate pathways that may be taken by egos, alters, and third parties that can explain actors’ network mobility, their locations in their networks, and ultimately, the size and shape of the networks to which they belong.

## 3. Alters’ Aversion

### 3.1. Study 1

As we noted above, those who are higher in neuroticism are characterized by their negative emotions as well as by their displays of challenging interpersonal behavior [12,20]. Consequently, people will be averse to initiating ties with HN actors relative to actors low in neuroticism (LN). The reluctance of others to reach out and initiate ties with HN actors will sharply constrain these actors’ opportunities to become well connected. For this, and all following studies, all subjects gave their informed consent for inclusion before they participated in the study. The study was conducted in accordance with the IRB (approval number: 1101001906).

**H1.** People will express relatively low intentions to initiate a tie with an HN actor compared to an LN actor.

### 3.2. Method

#### 3.2.1. Participants and Design

We recruited participants and conducted the study using Amazon Mechanical Turk (AMT). AMT is an online labor market that is used by experimentalists as a source of experimental data [31]. Comparisons of laboratory data and participants [32,33,34,35] indicate that the AMT samples are more representative than traditional college samples, and the data are at least as reliable. Results from experiments conducted via AMT were consistent with those collected in the lab of a Midwestern university, and collected on Internet boards [36]. 119 participants volunteered. 61 (51.3%) were men and 57 (47.9%) were women (1 did not report his/her sex). The mean age was 36.65 years (*sd* = 13.34), and the range was 19 to 70. Our software tracked participant IP address and Mechanical Turk Worker ID, allowing us to ensure that no individual participated more than once for any of the online studies.

The study was a 2 (participant sex: male, female) × 2 (target sex: male, female) × 2 (target neuroticism: lower, higher) between-subjects design. As has been done in prior studies of trait perceptions ([16], participants read a description of a target, here Mary or Mark. We embedded the manipulations of target sex and neuroticism in these descriptions. For a subset of the participants (*n* = 60), the target’s name was Mark, and for the remaining sub-sample the target’s name was Mary (*n* = 59). We assigned participants to one of two target neuroticism conditions (lower neuroticism, higher neuroticism). Sixty participants in the HN target condition read the following description:

Mary (Mark) is someone who enjoys reality television, and action movies. Most days, Mary (Mark) takes the train to work. When she (he) has a free weekend, she (he) enjoys taking short trips. Generally, Mary (Mark) is not considered someone who is calm and even-tempered. She (He) is frequently anxious or depressed, and is rarely positive and upbeat. Mary (Mark) tends to panic in tense situations, and is easily upset.
Participants assigned to LN target condition (*n* = 59) read:

Mary (Mark) is someone who enjoys reality television, and action movies. Most days, Mary (Mark) takes the train to work. When she (he) has a free weekend, she (he) enjoys taking short trips. Generally, Mary (Mark) is considered someone who is calm and even-tempered. She (He) is frequently positive and upbeat, and is rarely anxious or depressed. Mary (Mark) tends to be calm in tense situations, and is not easily upset.

#### 3.2.2. Dependent Variable Measures

As a manipulation check on target neuroticism, participants evaluated the extent to which the target was (1) calm, and (2) positive (1 = *not at all*, 7 = *extremely*). We averaged responses to these items to create a variable we could use to test the effectiveness of the target neuroticism manipulation (α = 0.95). We averaged participants’ responses to three (7-point) items to create an index of participants’ intentions to form a social tie: (1) If Mary/Mark sat next to you on the train, how likely is it that you might speak to her/him; (2) If Mary’s/Mark’s office were down the hall from yours, how likely is it you would stop by to chat; and, (3) If you had time to have lunch, how likely is it you would invite Mary/Mark to join you? (α = 0.90).

### 3.3. Results and Discussion

Target neuroticism was manipulated effectively, *F* (1, 119) = 144, *p* = 0.0001. Participants in the LN target condition judged Mary/Mark to be lower in neuroticism (*M* = 2.58, *sd* = 1.81) than did those in the HN target condition (*M* = 5.94, *sd* = 1.14).

We tested our hypothesis that people would be less inclined to initiate a tie with an HN target over an LN target by running a 2 (participant sex) x 2 (target sex) x 2 (target neuroticism) ANOVA. As we predicted, the only variable to have a significant effect on intentions to initiate tie formation was target neuroticism: participants were less inclined to initiate a social tie with an HN target (*M* = 3.26, *SD* = 0.18) than with an LN target (*M* = 5.48, *SD* = 0.18), *F* (1, 118) = 74.80, *p* = 0.0001; Table 1, Study 1). Neither the target’s sex nor the participants’ sex had any impact on the tie initiation decisions.

Potential alters showed little interest in forming social ties with HN targets, as predicted, providing necessary ‘proof of concept’ to allow us to move forward. Alters’ disinterest means that HN actors stand a relatively lower chance of forming the ties necessary to become well-connected in their networks. While indicative, there are two specific limitations to Study 1 that we address in our second experiment. First, rather than focus on behavioral intentions as we do in Study 1, we measure participants’ behaviors directly by having them choose people to occupy a particularly well-connected network position. Second, we also investigate the cognitive mechanisms that may explain people’s choice of LN over HN actors.

## 4. Study 2

Choosing not to initiate a tie is one way in which potential alters can prevent HN actors from becoming better connected. Another means is by withholding opportunities for actors to connect with third parties. For example, take the case of a manager who assigns associates to important client teams or to visible campaigns. Being appointed to better-connected spots can move people into more central network positions. However, if managers find HN employees aversive or if they believe that others will, managers may block them from these opportunities. Over time, being overlooked this way this will move HN actors to peripheral spots.

**H2.** Decision makers will assign HN actors to relatively less well-connected positions in networks.

In our review above, we noted the behavioral correlates of neuroticism that can explain why others may avoid actors high in this trait—their negative affect and vulnerability to stress, for instance [37]. Here we test the cognitive mechanisms that provide explanation by focusing on the schemas others hold about HN actors and their suitability for particular locations.

According to the stereotype content model [38,39], group stereotypes from along two dimensions: warmth and competence. Judgments of others’ trustworthiness, friendliness, likeability, and kindness comprise assessments of warmth. Competence judgments are made based on assessments of a group’s or person’s skill, intelligence, and efficacy. Studies linking stereotypic judgments to the Big Five indicate that warmth is strongly and positively associated with extraversion whereas competence perceptions show similarly strong and positive connections to conscientiousness [40,41]. Importantly for our work, neither warmth nor competence is positively associated with neuroticism. Unlikely to be judged as either particularly warm or competent, actors scoring highly on neuroticism are also unlikely to be nominated for central spots in networks. 

**H3a.** Decision makers’ judgments of HN actors’ (low) warmth will mediate the link between neuroticism and being chosen for a central position.

**H3b.** Decision makers’ judgments of HN actors’ (low) competence will mediate the link between neuroticism and being chosen for a central position.

### 4.1. Method

#### 4.1.1. Participants, Procedure and Design

We recruited 98 participants (51 men and 47 women) from our school’s experiment database. Each participant received 1 laboratory credit for participating.

The study was a 2 (participant sex: male, female) × 2 (candidate sex: male, female) × 2 (candidate neuroticism: lower, higher) between-subjects design. We told participants that they would be selecting another participant for a position on a team that would work together on an activity in the same laboratory the following week.

We described the team’s task as a four-person collaboration to produce a page for a student-run website in Figure 2. Participants saw a graphic depiction of the four positions teammates could hold in the group and the lines of communication allowed between people in each of the positions.

While a stylized network, the team we depicted shows differences in the structural centrality between those in the B and C positions (connected to two others—two ties) and those in the A and D positions (connected to one other—one tie).

Before participants arrived in the laboratory, we created a packet of information about a fictitious candidate who was intended to be a prospective teammate for a subsequent experiment. Embedded in the packet was the manipulation of the candidate’s trait-level neuroticism. Across conditions, the GPA (3.54), SAT score (1410), age (20), and work experience of the candidate was identical. Profiles also included three job-related handwritten weaknesses that were ostensibly offered by the candidate, as well as the candidate’s Big Five neuroticism score (relative to scores on other traits). These last two varied across conditions, and served as our manipulation of neuroticism. The LN candidate described her weaknesses thus: 1) “I’m pretty relaxed, even when things are crazy, “2) “Because I don’t freak out, sometimes people see this as me not caring about the work,” and 3) “My catering supervisor used to tell me to smile more at work.” The candidate’s scores on the non-neuroticism Big Five items were moderately high, and the neuroticism score was low (1.80 out of 7). We provided the meaning of these scores, along with the definitions of the traits.

In the HN condition, the candidate’s written description of three job-related weaknesses, included: 1) “I tend to freak out when anything stressful happens,” 2) “I know that I can get kind of irritable, and sometimes I take it out on other people,” and, 3) “My catering supervisor used to tell me to smile more at work.” This candidate’s score on the neuroticism measure was high (6.20). After reading the packet, all participants assigned the candidate to one of four positions in the work group, and answered questions related to this decision.

#### 4.1.2. Measures

We checked the manipulation of candidate neuroticism using a mean of two 7-point items: “To what extent is the participant you’ve read about: neurotic, stressed” (*α* = 0.95). We also measured participants’ judgments of the candidate’s competence (effective, capable: *α* = 0.72) and warmth (friendly, likeable: *α* = 0.87). Participants’ choice of network position for the candidate served as our primary dependent variable. Positions A and D were the less-connected positions, whereas B and C were the better-connected spots. Structurally, B and C are the same nodes within the network, as were A and D, and thus were considered the same choice.

### 4.2. Results

Candidates in the HN condition were perceived to be higher in neuroticism (*M* = 6.04, *sd* = 0.61) than those in the LN condition (*M* = 2.04, *sd* = 0.91), *F* (1, 96) = 657, *p* = 0.0001, making the manipulation effective. As predicted, HN candidates were significantly less likely to be assigned to a better-connected position in the network compared to LN candidates who were more likely to be assigned to better-connected positions (see Figure 3), χ^2^ (98) = 25.80, *p* = 0.000.

To explain participants’ decisions, we compared their perceptions of candidates on warmth and competence dimensions. As predicted, HN candidates were judged to be both significantly less warm (*M* = 3.54, *sd* = 0.88) and less competent (*M* = 4.36, *sd* = 0.92) than LN candidates (warmth: *M* = 5.13, *sd* = 0.97) (competence: *M* = 5.11, *sd* = 1.11), competence: *F* (1, 96) = 13.34, *p* = 0.0001; warmth: *F* (1, 96) = 70, *p* = 0.0001; Table 1, Study 12).

To test whether these perceptions explain the assignment of candidates to positions, we ran a set of regressions, in which the dependent variable was dichotomous [42]. Candidate neuroticism positively predicted assignment to the less-connected position (*b* = −1.17, *p* = 0.0001). When we included both neuroticism and competence judgments as predictors, both remained significant (neuroticism: *b* = −1.04, *p* = 0.000; competence: *b* = 0.89, *p* = 0.01), indicating no mediating effect for competence judgments.

When both warmth and neuroticism were included in the regression, neuroticism fell to non-significance, and only warmth predicted assignment (*b* = 1.19, *p* = 0.000). Despite the fact that HN candidates are seen as relatively less competent, it is their low warmth that interfered with participants’ assignments to better-connected network positions. Seeing HN actors as not particularly likeable, and thus assigning them to less central network spots is a potential explanation for Klein and colleagues’ [7]) observation that over time HN actors move into more peripheral positions.

Across two different samples and relying on two different kinds of network activity, we found support for the aversion hypothesis. Potential alters were disinclined to initiate ties with HN targets, and in the second study, a candidate’s neuroticism loomed large in alters’ appointment decisions. Though HN candidates were judged to be relatively less competent, this judgment did not explain alters’ decisions. Rather, it was their relatively low likeability that interfered with the assignments. With fewer opportunities to accept overtures, our results suggest that HN actors stand a relatively lower chance to occupy better-connected network positions, and this may explain why they find themselves in peripheral spots.

In our model we also hypothesized that actors have agency, choosing to pursue particular positions in social networks, or not. In the case of HN actors who occupy more peripheral network positions over time [7], we refer to this as avoidance, and test it in two experiments.

## 5. Actor’s Avoidance

As we have noted, HN actors are prone to anxiety, and to find uncertain situations especially challenging [18,19]. Given their relative difficulty interacting with other people, as well as the uncertain costs and benefits associated with being well connected, we posit that better-connected spots will be less appealing for HN actors compared to peripheral positions with fewer ties and obligations. Thus, one reason HN actors are likely to be found on the periphery may be because they find better-connected, more powerful positions unappealing, discouraging them from selecting them. We test this avoidance hypothesis in Studies 3 and 4.

### 5.1. Study 3

Social networks researchers commonly study two kinds of network structures—centrality networks and brokerage networks. Centrality networks can be thought of as comprising hubs and spokes, with the central actor being the hub [43,44,45]. Here, the social capital the central actor accrues comes in the form of social and emotional support as well as reputational benefits. While centrality networks are rich in the types of social capital sought after for personal well-being, some argue that they lack the characteristics that allow actors to access other kinds of resources [46,47,48]. Brokerage networks [47,48,49], in contrast, comprise separated clusters of networks with minimal ties connecting them. Actors here derive social capital through their ability to occupy the space between the clusters, gaining resources by being the sole conduit between groups.

Both kinds of networks place demands for interaction on those who occupy the powerful positions. With more ties to tend in the case of centrality networks or relationships to disparate groups to maintain in brokerage networks, well-placed players must devote resources to nurturing their relationships or risk weakening their ties. For all the documented benefit to occupying central and brokerage positions, respectively, maintaining them carries costs.

One cost comes in the form of uncertainty. At the outset, actors typically do not know how much of their resources will be necessary to build and maintain ties. In fact, how much benefit will flow from newly-created ties is also unknown. This is likely to loom large for among those who are higher in neuroticism whose sensitivity to uncertainty is well established [18]. By this logic, we expect powerful positions—whether central or brokerage spots—will be relatively unappealing to these actors. Moreover, we expect the costs of connectedness will be especially salient for them.

**H4.** Neuroticism will be positively related to judgments that well-connected positions carry costs.

**H5.** Neuroticism will be negatively related to preferences for well-connected positions.

### 5.2. Methods

#### Participants and Design

We recruited participants and conducted the study using AMT. Two hundred and sixty people volunteered to participate. 141 (54.2%) were men and 118 (45.4%) were women (1 did not report his/her sex). The mean age was 32.05 years (*sd* = 11.7), and the range was 18 to 78.

Participants completed an abbreviated form of the International Personality Item Pool (IPIP) to allow us to measure personality [50]. The 25-item instrument included five items for each factor of the five-factor model. Participants rated how much they agreed with each item on a five-point scale (1 = *“strongly disagree,”* 5 = *“strongly agree.”*). Focusing on the neurotics subscale, we combined the five items to create a neuroticism index (α = 0.69).

To measure respondents’ interest in taking on central roles in their networks, we asked, “In your normal everyday life, how interested are you in gaining positions that are central?” (1 = *not at all interested*, 7 = *extremely interested*). We also asked participants three questions to gauge their perceptions of the costliness of well-connected positions: 1) To what extent do you believe being well-connected to others brings costs? 2) To what extent do you believe being well-connected places a demand on you emotionally? 3) To what extent do you believe being well connected places a large demand on you cognitively or intellectually? (α = 0.69).

### 5.3. Results

When asked about their preference for more or less central positions in a hypothetical network, participants in the sample were nearly evenly split (preferred less central position = 121; preferred more central position = 134). Those who preferred a more central position also understood the value of this position, reporting that it was important to be seen as central in work and friendship networks (*M* = 4.89, *sd* = 1.46) compared to those who preferred a less central position (*M* = 4.36, *sd* = 1.57), *F* (1, 195) = 27.66, *p* = 0.0001. This suggests that respondents who prefer central roles are particularly sensitive to the tactical advantages afforded by these positions.

Turning to our hypotheses, we postulated that neuroticism would be positively related to seeing better-connected positions in networks as costly, and this was the case: *b* = 0.15, *t*(188) = 2.12, *p* < 0.05. We also expected that neuroticism would undercut people’s interest in taking on better-connected network positions (H5). Indeed, this was the case, *b* = −0.20, *t*(246)= −3.24, *p* < 0.001. The greater one’s neuroticism score, the less interested one was in well-connected positions. These findings lend support for our argument that HN individuals find better-connected positions relatively less appealing. However, as was true in Study 1, this study measures intentions rather than actual behaviors. Moreover, people only indicated their preferences for being “central” in a network. Whether neuroticism shapes preferences in both centrality and brokerage networks remains to be seen.

## 6. Study 4

Whereas in Study 3 people reported their interest in occupying positions defined generally as “central,” in Study 4 we took care to spell out what central actually looked like. For people presented with a centrality network, we measured their choice between a more centrally-located position or less centrally-located one. For those presented with a brokerage network, we measured whether their choice was a brokerage role or not. We had no theoretical reason to hypothesize that neuroticism would lead people to view powerful positions in centrality and brokerage networks differently. Rather, we hypothesize simply that neuroticism will be negatively related to preferences for powerful positions, regardless of network type.

**H6.** Neuroticism will be negatively associated with choosing more powerful positions in both centrality and brokerage networks.

### 6.1. Method

#### 6.1.1. Participants, Procedure and Design

We recruited participants from a university experiment database. Ninety eight participants took part; 53 (54.1%) were men and 45 (45.9%) were women. The mean age was 20.83 years (*sd* = 3.66). Each participant received 1 laboratory credit for participating.

We told participants that they would take part in a team task later in the semester, and that their task today was to select their preferred position on that team. After reading about the team’s task—creating a blog for a student-run university website—they viewed one of two networks. Half the sample chose a position in a brokerage network (Figure 4), and the other half chose a position in the centrality network (Figure 5).

In the brokerage network, the K position is the broker position, as it is low in network constraint, and according to Burt [47,48] or Granovetter [49], the most powerful position. In the centrality network, the C position is the central position, and the more powerful position [43,44,51,52,53,54].

#### 6.1.2. Measures

In the centrality network, we coded participants’ choice as 1 if they selected the C position, and 0 if they chose another position. For those who viewed the brokerage network, we coded their choice as 1 if they selected the K position, and 0 if they chose any other position. This was our primary dependent measure. In addition, however, we measured participants’ scores on a neuroticism scale, as we had done in Study 3 (α = 0.78).

### 6.2. Results

To test our hypothesis that neuroticism would be negatively related to choice of powerful network positions, we estimated a binary logistic regression using position choice as the dependent variable and neuroticism score as the independent variable. As we expected, neuroticism negatively predicted participants’ choice of power positions, *b* = −.45, *p* < 0.05; odds-ratio 0.64, regardless of network type (or participant gender).

Together, Studies 3 and 4 provide evidence that more powerful network spots are less appealing the higher the decision maker’s neuroticism score. This is true regardless of what defines the powerful position—brokerage or centrality. Results suggest that a preference for avoiding powerful positions may explain why neuroticism is associated with more peripheral network spots.

## 7. General Discussion

Work on the role of actors in networks tends to address questions of who—who are the people who occupy particularly advantageous spots, and what characteristics do they share? Answers point to a variety of traits that make the difference—self-monitoring, Big Five personality traits, self-esteem, among others. Largely unexplored, is the question of *how* people with particular characteristics come to occupy various positions has been unexplored, likely due to a lack of experimental control necessary for causal tests. Yet, understanding and predicting social network evolution is difficult without better insight into the cognitive and behavioral mechanisms that can explain how actors move to particular network positions. After all, traits indicate preferences for particular perceptions, emotional expressions and behaviors. And relying on experimental methods, researchers can explain how traits come to predict network outcomes [4,5,47,48,51,52].

We started with the finding that actors who are high in trait-level neuroticism are likely to move into more peripheral network spots over time. Drawing on growing psychological literature on the cognitive and behavioral correlates of neuroticism, we conceptualized a model that offered two distinct accounts: (1) alters’ aversion, and (2) actors’ avoidance. As predicted, results of two experiments showed that potential alters find HN actors *aversive*, making these actors less appealing targets for tie initiation and for recommendations to central positions, largely due to their perceived lack of likeability. These results indicate that alters have the potential to exert powerful effects on actors’ network mobility, and ultimately, their network position. In the specific case of HN actors, potential alters do this by constraining their opportunities to build the ties—and the social capital—necessary to become powerful players in their networks. Fewer opportunities mean more peripheral positions, which in turn, provide fewer opportunities. Our findings suggest that the process of network evolution is recursive and reinforcing.

If we had ended our investigation here, we would have concluded that alters’ aversion to HN actors is responsible for these actors’ place in their networks. However, we also considered the role that HN actors themselves play, proposing an *avoidance* hypothesis. Owing to their relative discomfort in social situations, we postulated that these actors would find well-connected positions unappealing. Indeed, the greater the actor’s neuroticism, the less likely she was to select better-connected positions in teams when given the chance, and this was true in two different kinds of networks. Analyses revealed no differences in perceptions of benefits associated with being connected, but neuroticism did affect perceptions of costs. In brokerage networks, neuroticism was linked to preferences and choices for less well-connected positions. The implication is that neuroticism may be undermining actors’ opportunities to acquire and transfer valuable resources. Again, avoidance may explain why neuroticism is associated with peripheral network positions.

## 8. Theoretical and Practical Implications

Just as it takes two to tango, it takes two to build a network tie. Observing an actor moving through a network is to witness an actor building ties with some people, and not others. At its core, network mobility is fundamentally about tie formation. Actors and alters initiate and accept—or avoid and decline—overtures, which directly affect an actor’s place in her network. Moreover, alters can affect whether actors connect with third parties, too, and thus exert a powerful influence over actors’ networks. We examined these processes directly, making explicit tie formation via assumptions about behavior as a way of accounting for links between traits and network position. As noted earlier, actor level traits do not impact network evolution—rather, traits drive behavioral expressions or beliefs about likely behaviors and these beliefs in turn drive network churn [55,56].

By investigating the cognitions and behaviors of both actors and alters, we acknowledged the roles played by both in shaping people’s network outcomes. This approach marks a departure in the literature, as much of the work centers on the role of agentic actors [57]. Here we recognize and empirically test how actors and alters can provide and constrain opportunity to shape network evolution.

Although we started this section by arguing that building ties is critical to network evolution, our results suggest a modification to this point. Decisions not to initiate a tie or, in the case of actors, to choose to initiate fewer ties rather than more, may be just as important as building ties for network evolution (regarding firm level dyadic ties, see [58]). Taking this point further, it seems that while it takes two to build a tie, it takes just one to avoid it. While it is certainly understandable that the majority of network research focuses on tie formation, decisions to decline an invitation, or to pass on an opportunity, are just as important. From a professional and practical standpoint, it is likely that HN actors are less likely to self-select into careers or positions characterized by centrality demands in either a Burt or Coleman fashion. Further, it is unlikely HN actors are given opportunities to inhabit such networks by persons in positions of power, owing to the beliefs and assumptions of said person. From a social and personal standpoint, this research suggests that HN actors are also unlikely to occupy network positions ripe in emotional support, or with access to advice and counsel. Somewhat ironically, then, HN actors—who arguably require more, not less, practice in engaging in interpersonal communication—are likely to receive less over time.

In our studies, some people—here, actors higher in trait-level neuroticism—systematically avoided spots rich in social capital opportunity. In fact, in Study 3, nearly half our sample expressed no interest in occupying what would be considered better-capitalized network positions. This was not because they failed to recognize the benefits inherent in these spots, because they did. Rather, they saw these positions as imposing high costs. What appears to be a cost-benefit calculation led some to concede “better” positions to other actors. Un-spanned structural holes may exist because the demands associated with bridging are seen as outweighing the potential benefit. Indeed, this would help explain why brokerage is so short lived [47,48].

Whereas HN actors showed relatively little interest in occupying advantageous positions in general, it was not the case that all HN actors felt this way. For those who preferred the better-connected position, our work offers some practical advice. Given the chance to interact, HN actors may be able to overcome potential alters’ biases against them, breaking down an obstacle to moving into better position. These efforts will be more likely to succeed if HN actors are aware of the behaviors that tend to set them apart from others, and if they develop a wider repertoire of behaviors. Presenting a more agreeable, calmer demeanor should enable them to move along to better positions in the network, if that is what they want.

Construing better-connected positions as long on costs and short on benefits will keep HN actors out of positions that offer desirable resources. While the alter component of our research suggests that HN actors should work towards managing how others view them, the actor half suggests that it behooves HN actors to reconceptualize connectedness as something useful and desirable. Nearly half our sample noted the costs of connectedness, only HN actors consistently avoided taking on these spots.

Our results show the value of experimental methods and psychological theory in explaining the shifts in shape and structure of networks. Social networks evolve as a consequence of social cognition and social action, making their study amenable to the application of social psychological methods and theories. Moreover, the controls experiments provide enable a precise focus on specific cognitions and behaviors and establish causal stories regarding network development.

## 9. Limitations and Future Directions

To provide the controls necessary, we described HN actors rather than have HN actors interact with potential alters. Our descriptions may have made neuroticism particularly salient while downplaying other equally or more important traits or attributes. At the same time, it could be the case that interacting with HN actors makes their neuroticism even more salient than it is when the participant reads about them. Future work should investigate the factors that might mitigate the importance of trait-level neuroticism in alters’ decision making. Recent work, for instance, shows that characteristics like physical attractiveness are powerful for drawing alters in and having a network assemble around ego [59]. Other work demonstrates that those high in self-monitoring are often able to adapt to social situations, and adopt new behaviors or mask undesirable ones [5].

Differences in how people construe network “opportunities” may explain people’s spots in networks. As we noted, we found that people see the same opportunity in a particular network location, but they view the costs much differently, leading some to seize it while others do not. More work should be done to understand what is at the root of these perceptual differences. Do some erroneously under- or overestimate the costs? Or do people have an accurate read on the costs, but vary in the resources they have to manage them?

We showed that both aversion and avoidance processes potentially can explain why neuroticism is associated with peripheral positions. However, our data did not allow us to say which process exerted a more powerful effect. For other traits or characteristics, however, the relative influence of that characteristic on alters or egos may be more easily determined. Actors who are low in agreeableness, for instance, may be keen to move into powerful spots, but aversion on the part of others may prevent this, leaving them in less central positions. Further work should continue to explore the relative strength of the influence of alters and actors on network formation and evolution.

Our findings also suggest that behaviors on the part of ego and alter systematically may be pushing actors who score highly on neuroticism to the network periphery, regardless of network configuration (broker v centrality network). This, in turn, suggests that particularly salient personality characteristics such as neuroticism may exhibit signs of homophily [60]. This tendency toward homophily may be tempered if even HN actors find one another to unappealing interaction partners. If HN actors also relegate one another to network peripheries or refuse tie formation overtures, this suggests that HN actors are continually finding new and undiscovered network peripheries—ultimately suggesting a very nomadic social and professional existence.

In these studies, alter is making decisions which result in network marginalization of ego, and ego is making decisions which result in network marginalization of herself. Because our conceptual model treats alter and ego decisions as separate decision trees, we do not test which set of decisions exerts a stronger impact.

Whereas we focused on the role of trait-level neuroticism, other traits also may make a difference to alters’ and egos’ preferences, decisions, and actions. high levels of openness to experience, for example, may encourage some actors to be at higher risk for building ties to demographically diverse nodes in distant clusters.

Additionally, further longitudinal data is needed to more fully address questions regarding network change over time. While we have evidence that our proposed mechanisms lead to a particular type of network outcome, we do not have observational evidence which more fully details and elaborates on this process. Longitudinal studies with repeated observations of actor movement within networks would contribute greatly to the existing literature on individual differences and their role in the development of networks.

Last, it is important to consider context. Here, we tested the impact of neuroticism on actor location within work groups. That is, a group with semi-predetermined network and social roles, and with a goal or shared purpose. What our results cannot tell us is how ego and alter will perceive opportunities and each other in non-work settings.

## 10. Conclusions

Advances in computing and modeling capabilities have revealed how dynamic social networks can be [3,61]. This work provides important descriptions of network evolutionary patterns, but it raises new questions about the precise mechanisms driving at may explain them. Conceptualizing network evolution as an outcome of the preferences, decisions, and actions of both actors and alters, there is a tremendous opportunity for micro-level researchers to bring their theories and methods to bear to explain network evolution. Further, understanding network evolution as a function of not only the ties we make and maintain, but as a function of the ties we do not accept or that we allow to degrade, allows a more nuanced perspective on network development. Indeed, the absence of ties within a network may say as much, or more, as the presence of ties within a network.

## Figures and Tables

**Figure 1 behavsci-09-00069-f001:**
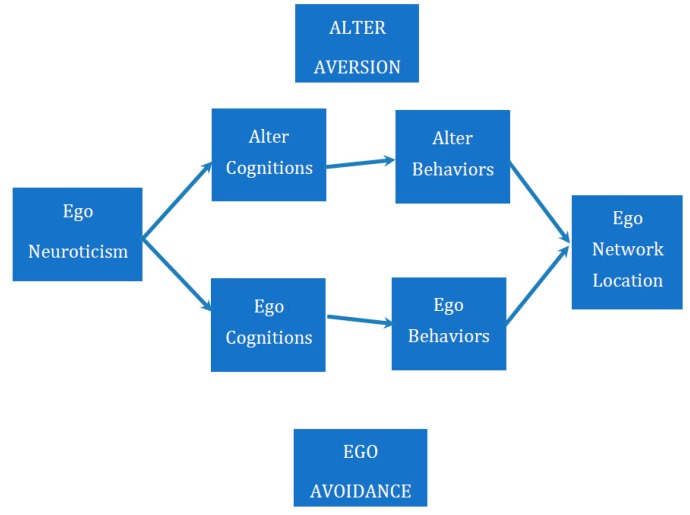
Conceptual model of ego’s and alter’s roles in determining actor network position.

**Figure 2 behavsci-09-00069-f002:**
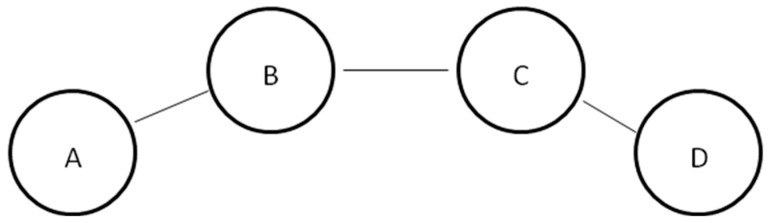
Study 2—Depiction of the lines of communication between members of the four-person work team.

**Figure 3 behavsci-09-00069-f003:**
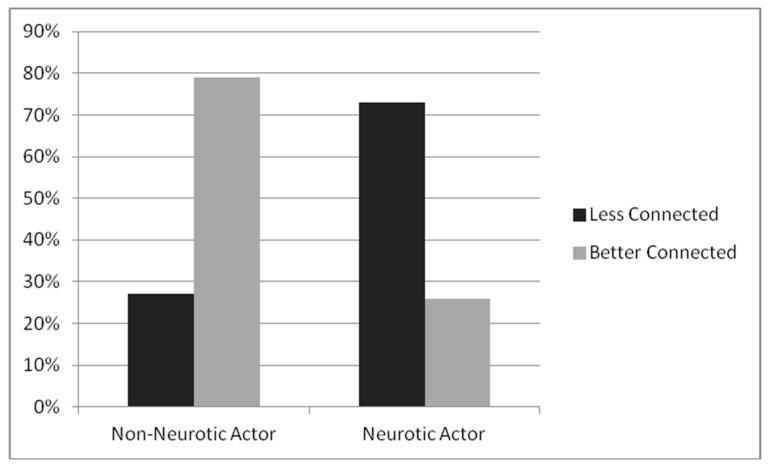
Study 2—Percentage of candidates assigned to less-connected and better-connected positions.

**Figure 4 behavsci-09-00069-f004:**
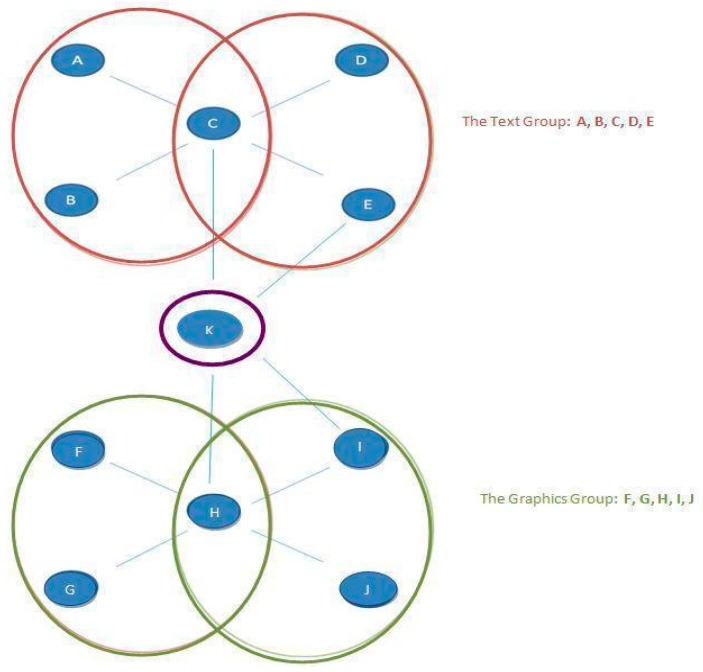
Study 4—Depiction of a brokerage network.

**Figure 5 behavsci-09-00069-f005:**
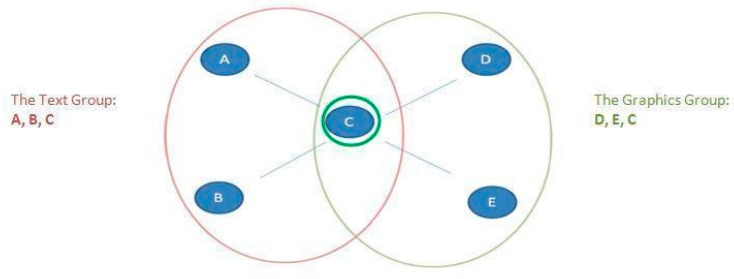
Study 4—Depiction of a centrality network.

**Table 1 behavsci-09-00069-t001:** Summary of Study 1 and Study 2 means and standard deviations.

**Study 1**		
	**LN Condition**	**HN Condition**
Participant Perception of Target Neuroticism	M = 2.58	M = 5.94
	SD = 1.81	SD = 1.14
Participant Intention to Initiate Tie With Target	M = 5.48	M = 3.26
	SD = 0.18	SD = 0.18
**Study 2**		
	**LN Condition**	**HN Condition**
Participant Perceptions of Target Neuroticism	M = 2.04	M = 6.04
	SD = 0.91	SD = 0.61
Participant Perceptions of Target Warmth	M = 5.13	M = 3.54
	SD = 0.97	SD = 0.88
Participant Perceptions of Target Competence	M = 5.11	M = 4.36
	SD = 1.11	SD = 0.92

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
