# Peer review of "The Push and Pull of Network Mobility: How Those High in Trait-Level Neuroticism Can Come to Occupy Peripheral Network Positions"

_behavsci, 2019, doi:10.3390/bs9070069_

Round 1
Reviewer 1 Report
Strengths of the Paper/Study
This paper is very well written, clear, and organized. The four studies are well-designed to accurately target both decisions and actions taken that center on neuroticism (i.e., high validity). The studies align with the hypotheses discussed. They are presented in a logical fashion. The data are interesting and are presented in a way that is easy for readers to understand.
Broad Critiques
My main issues I had with the paper have to deal with how the paper is framed, and what the data can actually tell us.
The abstract is very accurate in summarizing the theory and the data. However, parts of the introduction set out some fairly lofty and ambitious goals around what it can tell readers about how actors with high neuroticism move outward within networks. It proposes two mechanisms by which this happens: 1) alters do not nominate HN actors for more central positions, and 2) HN actors actively do not choose to occupy those positions. From this, I think the main contributions of the paper should be framed as the following: 1) you offer support for the first phenomenon, which has been shown in previous work, and 2) you offer an additional mechanism whereby HN actors have some autonomy in this process and chose to occupy less central positions in the network.
My main critique, then, is that these data currently do not tell us about mechanisms through which HN actors move outwardfrom central positions to less central positions. Instead the current data provide evidence for why people with neuroticism occupy less central positions and perhaps may be less likely to move inward. While it does seem plausible that these same mechanisms could also explain potential outward movement in the network, future studies are needed to test for this using prospective data, or data that show HN actors are in central positions in the first place. As such, the way much of the introduction is written does not align with what the current study can actually tell us.
From your introduction:
“Who are the people becoming more deeply embedded in clusters, and who are the ones moving out to the periphery? Exactly what are the decisions and actions that drive actors’ mobility in their networks? And whose decisions matter the most: are network dynamics affected by the ego, the alter, or some combination of the two?
Your data do not answer many of these questions. For example, you test HN versus LN, but we do not know enough about other types of personality traits and network dynamics to answer broader questions around “who are the people” moving in and out of network clusters. Additionally, you do not test whose decisions matter most, only that both the ego and the alter make decisions and act in ways that seem to matter when tested in separatestudies/models. I would reframe this section, because as such, it sets the data up to answer broader questions than it actually can.
A second critique on framing is to clarify what you mean by ‘networks’, and to define which type of networks you are referring to. Professional networks? Familial/personal networks? For example, would this theory hold for how family members treat people with neuroticism? Or close friends?
Finer Points
Again, the paper is very well written and does a good job of referring to individuals with high levels of neuroticism until one of the paragraphs in the discussion section (lines 594-604), where the authors revert to using the term ‘neurotics’ repeatedly to refer to people with HN. I would avoid this language and stick with the phrasing used throughout the rest of the paper.
Use the title to convey the findings of this paper, avoid language that labels people ‘the neurotic’.
Line 22-23 in abstract is a fragmented sentence.
Study 1 examines how likely you would be to engage with a strangerwho is neurotic, not how you might treat someone who is already embedded in the network. You do point out that this study is fairly limited in its ability to tell us about how people who know one another might treat each other.
Study 2 shows that colleagues would place HN folks in less connected positions and you then state that this is possible evidence for why people high in neuroticism would occupy “peripheral positions in theirnetworks”. However, this reasoning seems problematic, especially since it is hard to argue that an ego occupies a peripheral position within their own network, without more information about the extended network.
In study 3 you note that HN people believe highly central positions are too emotionally taxing. But you do not comment on whether HN people feel that it is important to be in centrally connected positions. This would be interesting to know.
Study 4 results: Please present results as odd ratios for binary logistic models. Is this estimate (b= -.45) the exact same for both types of networks? Do these logit models control for gender?
Lines 642-644: “Our findings also suggest that behaviors on the part of ego and alter systematically push HN actors to the network periphery, regardless of network configuration.” This line is both problematic and confusing since the rest of the paragraph does not elaborate on this point. The data do not show that HN people are pushed, nor pushed outward. It does show that HN people are placedinto less peripheral positions and may be less likely to move inward. What is meant by ‘regardless of network configurations’?
On lines 29-31 you state: “Indeed, our goal is to provide causal evidence of the alter and ego processes which result in those high in trait-level neuroticism being peripheral network players.” However, there is no causal evidence or data presented. Do you mean that the goal of the paper is to provide potential causal pathways, or mechanisms, by which HN actors come to occupy peripheral positions within the network? Please reframe the goal.
Would the use of tables to present the data be helpful for readers to quickly understand the main points of your paper?
Author Response
This paper is very well written, clear, and organized. The four studies are well-designed to accurately target both decisions and actions taken that center on neuroticism (i.e., high validity). The studies align with the hypotheses discussed. They are presented in a logical fashion. The data are interesting and are presented in a way that is easy for readers to understand.
#Thank you!
Broad Critiques
My main issues I had with the paper have to deal with how the paper is framed, and what the data can actually tell us.
The abstract is very accurate in summarizing the theory and the data. However, parts of the introduction set out some fairly lofty and ambitious goals around what it can tell readers about how actors with high neuroticism move outward within networks. It proposes two mechanisms by which this happens: 1) alters do not nominate HN actors for more central positions, and 2) HN actors actively do not choose to occupy those positions. From this, I think the main contributions of the paper should be framed as the following: 1) you offer support for the first phenomenon, which has been shown in previous work, and 2) you offer an additional mechanism whereby HN actors have some autonomy in this process and chose to occupy less central positions in the network.
# We appreciate your comment here and agree that we have overstated our claims in the paper’s Introduction. We have revised the paper to reflect your point that we have empirical support for alters not nominating HN actors and that we have some suggestive results for the role that HN actors may play in their peripheral network locations.
My main critique, then, is that these data currently do not tell us about mechanisms through which HN actors move outwardfrom central positions to less central positions. Instead the current data provide evidence for why people with neuroticism occupy less central positions and perhaps may be less likely to move inward. While it does seem plausible that these same mechanisms could also explain potential outward movement in the network, future studies are needed to test for this using prospective data, or data that show HN actors are in central positions in the first place. As such, the way much of the introduction is written does not align with what the current study can actually tell us.
# You make a very good point here. The work of Klein and her colleagues shows that actors who score higher in neuroticism move out toward the periphery over time (3 months in their study). Our studies were meant to take that as starting point from which we proposed and tested mechanisms that could explain these effects.
You are correct that our results show how actors’ neuroticism can result in actors being in peripheral positions. We have made an effort to edit the manuscript to better reflect our results (eliminating any mention that our results show movement, but they help explain others’ findings of movement).
From your introduction:
“Who are the people becoming more deeply embedded in clusters, and who are the ones moving out to the periphery? Exactly what are the decisions and actions that drive actors’ mobility in their networks? And whose decisions matter the most: are network dynamics affected by the ego, the alter, or some combination of the two?
Your data do not answer many of these questions. For example, you test HN versus LN, but we do not know enough about other types of personality traits and network dynamics to answer broader questions around “who are the people” moving in and out of network clusters. Additionally, you do not test whose decisions matter most, only that both the ego and the alter make decisions and act in ways that seem to matter when tested in separate studies/models. I would reframe this section, because as such, it sets the data up to answer broader questions than it actually can.
# You are right that in this paper we focused only on trait-level neuroticism, and by manipulating a target’s neuroticism in Studies 1 and 2, we believe we can draw conclusions about how people think about and afford opportunities to these targets. Yes, we are only studying this trait. We have edited the paper to reflect your comments.
A second critique on framing is to clarify what you mean by ‘networks’, and to define which type of networks you are referring to. Professional networks? Familial/personal networks? For example, would this theory hold for how family members treat people with neuroticism? Or close friends?
# As our experiment descriptions indicate, we were interested in work-based networks (e.g., the Mark/Mary study, the choosing a spot on a work team study). However, this was not clear in our introduction. We have edited the text to be more specific (page 4). Moreover, we now note our limited scope of context in our “Limitations” section, beginning at the bottom of page 29.
Finer Points
Again, the paper is very well written and does a good job of referring to individuals with high levels of neuroticism until one of the paragraphs in the discussion section (lines 594-604), where the authors revert to using the term ‘neurotics’ repeatedly to refer to people with HN. I would avoid this language and stick with the phrasing used throughout the rest of the paper.
# Thank you. We agree and have edited the paper so that we use more appropriate language. We now refer to these actors either as “those high in trait level neuroticism” or as “HN” actors.
Use the title to convey the findings of this paper, avoid language that labels people ‘the neurotic’.
# We have made the change and hope you agree that the title better reflects our findings.
Line 22-23 in abstract is a fragmented sentence.
# Thank you for noting this. We have slightly altered the abstract to reflect this suggestion, as well as suggestions made above regarding our theoretical/practical claims.
Study 1 examines how likely you would be to engage with a strangerwho is neurotic, not how you might treat someone who is already embedded in the network. You do point out that this study is fairly limited in its ability to tell us about how people who know one another might treat each other.
# Excellent point. We now mention this in the last paragraph of our “Limitations” section (page 30). We have noted that our context, here, is one characterized by a lack of familiar interpersonal relations, and that future work can test how the mechanisms might operate when actors are known to one another.
Study 2 shows that colleagues would place HN folks in less connected positions and you then state that this is possible evidence for why people high in neuroticism would occupy “peripheral positions in theirnetworks”. However, this reasoning seems problematic, especially since it is hard to argue that an ego occupies a peripheral position within their own network, without more information about the extended network.
# Yes. We have modified our language to reflect that this suggests a mechanism through which ego can be placed on the periphery of a given network, not necessarily their own (page 16, second to last paragraph).
In study 3 you note that HN people believe highly central positions are too emotionally taxing. But you do not comment on whether HN people feel that it is important to be in centrally connected positions. This would be interesting to know.
# We did not assess their judgments of importance, but we tried to get at their judgments of the position by asking about its benefits. Still, you make an interesting observation about importance.
Study 4 results: Please present results as odd ratios for binary logistic models. Is this estimate (b= -.45) the exact same for both types of networks? Do these logit models control for gender?
# Good point--gender, and network type, had no impact and as such, we collapsed across these variables. We now note this on page 23, under “7.4 Results”. We also now include the odds-ratio in the Results section, on the back-end of the reporting of coefficient and significance value. We would like to point out that, traditionally, odds-ratio’s can be difficult for some to interpret, and as such, we leave the inclusion of the odds-ratio to the Editors judgment.
Lines 642-644: “Our findings also suggest that behaviors on the part of ego and alter systematically push HN actors to the network periphery, regardless of network configuration.” This line is both problematic and confusing since the rest of the paragraph does not elaborate on this point. The data do not show that HN people are pushed, nor pushed outward. It does show that HN people are placedinto less peripheral positions and may be less likely to move inward. What is meant by ‘regardless of network configurations’?
# Thank you for pointing this out. Throughout the paper, we have modified our language to better capture our results. We also have noted, briefly, what we mean by type of network configuration (broker v centrality network). We hope this clarifies.
On lines 29-31 you state: “Indeed, our goal is to provide causal evidence of the alter and ego processes which result in those high in trait-level neuroticism being peripheral network players.” However, there is no causal evidence or data presented. Do you mean that the goal of the paper is to provide potential causal pathways, or mechanisms, by which HN actors come to occupy peripheral positions within the network? Please reframe the goal.
# Yes. Thank you. As you say, our goal was to provide evidence of potential causal pathways. We hope that our edits now reflect our intentions and results more clearly.
Would the use of tables to present the data be helpful for readers to quickly understand the main points of your paper?
# Thanks for bringing this up. We feel that a summary of pertinent means, etc, across Studies 1 and 2 would help the reader more quickly understand some of the main points of the paper. We now include a brief table on page 43, as well as make reference to it in the “Results” sections of Study 1, and Study 2.

Reviewer 2 Report
This research tackles the hypothetical association between neuroticism personality and network positions, which is a relatively underexplored topic with a lot of potentials to answer the question of the causality between network structure and personal agency. The study recruited respondents from Mechanical Turk and conducted four separate manipulated experiments to test their respective hypothesis.
Most of the problems arising presumedly from AMT do not bother this study due to its manipulated experiment nature, which is a strength. But please consider report the dropout rate, reliability, or other issues with the sampling.
Neuroticism in study 1 and 2 is defined and interpreted through descriptive narratives in vignette, the short vignette description does not fully covey all major points of neuroticism as formally defined by psychologists, please provide justifications or compare the vignettes with formal definitions, emphasizing their commonality.
In study 4, to say that actors with greater centrality constitute powerful positions is an idea from Coleman is questionable. Coleman just hadn't discussed his idea of social capital in the context of network centrality, he didn't have a chance to compare brokerage and centrality in the traditional sense of network theory either. Maybe add some justifications or cite others.
The authors have somewhat overstated the achievement of their study as laid out in the introduction. In the introduction, much of the gap and the needed breakthrough was about how the current scholarship could not use traditional survyes to answer the movement of actors in the networks and the evolution of full networks. However, the RC Experiments presented in this study were also conducted contempraneously to simply reveal how personality as manipulated could change people's preference for ties. This is a long stretch from the evolution of networks since we still lack the knowledge of the real time change of edges and nodes. In realistic settings, the formation and dissolution of ties depend on other mechanims than personality and preference, often stronger and more mechanic ones such as the exigency of existing ties, random process, etc. This self-forming process of any network cannot be explained by experiments like the current one. Please consider revise your discussions.
Author Response
This research tackles the hypothetical association between neuroticism personality and network positions, which is a relatively underexplored topic with a lot of potentials to answer the question of the causality between network structure and personal agency. The study recruited respondents from Mechanical Turk and conducted four separate manipulated experiments to test their respective hypothesis.
# Thank you.
Most of the problems arising presumedly from AMT do not bother this study due to its manipulated experiment nature, which is a strength. But please consider report the dropout rate, reliability, or other issues with the sampling.
# Thank you for commenting on this--our data only includes those who fully completed the experiment. We do not have observational data on those who failed to complete the task as their inputs are not retained. As we note in the footnote on page 9, though, completion rates, demographics among Turk Workers are similar to those encountered in the laboratory and in other traditional settings.
Neuroticism in study 1 and 2 is defined and interpreted through descriptive narratives in vignette, the short vignette description does not fully covey all major points of neuroticism as formally defined by psychologists, please provide justifications or compare the vignettes with formal definitions, emphasizing their commonality.
# You are correct that we did not capture the range of adjectives for neuroticism in our vignettes. We wanted to include the behaviors without drawing so much attention to them that participants could guess our hypotheses. We relied on Table 1 in McCrae & John’s 1992 paper where they provide a list six adjectives associated with neuroticism: anxious, self-pitying, tense, touchy, unstable, worrying. Moreover, we conducted a literature review to assess the affective correlates of neuroticism. That empirical work shows that people who are relatively higher in neuroticism react with more severe negative emotions, tend to more strongly react to recurring problems (the ‘neurotics’ cascade) and are highly susceptible to stress (e.g., Suls and Martin, 2005). In our short vignette, we hoped to capture the gist of these definitions and empirical findings.
In study 4, to say that actors with greater centrality constitute powerful positions is an idea from Coleman is questionable. Coleman just hadn't discussed his idea of social capital in the context of network centrality, he didn't have a chance to compare brokerage and centrality in the traditional sense of network theory either. Maybe add some justifications or cite others.
#We have now addressed this comment by including additional references. Beyond Coleman, many network scholars view centrality (as opposed to brokerage) as a potential source of interpersonal power, status, and influence. We appreciate this critique, though, as it caused us to include additional material which the reader may find useful. Please find new citations/references which point to the power of centrality in Study 4.
The authors have somewhat overstated the achievement of their study as laid out in the introduction. In the introduction, much of the gap and the needed breakthrough was about how the current scholarship could not use traditional survyes to answer the movement of actors in the networks and the evolution of full networks. However, the RC Experiments presented in this study were also conducted contempraneously to simply reveal how personality as manipulated could change people's preference for ties. This is a long stretch from the evolution of networks since we still lack the knowledge of the real time change of edges and nodes. In realistic settings, the formation and dissolution of ties depend on other mechanims than personality and preference, often stronger and more mechanic ones such as the exigency of existing ties, random process, etc. This self-forming process of any network cannot be explained by experiments like the current one. Please consider revise your discussions.
# Thank you for this comment. Both you and the other reviewer noticed this overstatement and we have edited the entire manuscript to make clear that, inspired by the work of Klein and her colleagues, we are exploring two mechanisms that could possibly explain Klein’s findings that actors higher in trait-level neuroticism move to the periphery of networks. We hope you agree that these edits better describe our results.
